# Factorized Gaussian Splatting for View-Inconsistent Degradations

## Abstract

Multi-view reconstruction often assumes cross-view appearance consistency, which is violated in the presence of participating medium and sensor artifacts. These view-inconsistent degradations induce distorted geometry and unreliable appearance in scene modeling. In this study, we propose Factorized Gaussian Splatting (F-GS), a unified scene modeling framework that explicitly decomposes the scene into three complementary components: Geometry, Medium, and Residual. The geometry component is stabilized by a lightweight view-aware surface gain and an optional screen-space normal-consistency prior to consolidate scene structure. The medium component is represented by sparse yet large Gaussians to model light attenuation through participating medium. The residual component captures view-dependent sensor noise via residual Gaussians, compensating for sensor-induced variations. This factorization prevents view-inconsistent degradations from contaminating geometry and appearance, enabling more accurate scene modeling. We instantiate and evaluate our F-GS in three representative regimes, thermal long-wave infrared, underwater, and foggy scene, where the view-inconsistencies naturally occur. Our F-GS significantly improves novel-view synthesis quality and geometric accuracy over other baselines. Code and Visualization are available at https://anonymous.4open.science/r/F-GS-83EE/README.md.

## 1 Introduction

For real-world sensing scenes, such as thermal, underwater, and foggy scans, precise modeling of both geometry and appearance is crucial for emerging applications in robotics, autonomous navigation, and immersive computing. State-of-the-art scene modeling approaches, including Neural Radiance Field (NeRF) Mildenhall et al. (2020) and 3D Gaussian Splatting (3DGS) Kerbl et al. (2023), recover scenes through novel view synthesis (NVS) and achieve high-quality modeling results on RGB imagery. However, these methods implicitly assume clean and cross-view-consistent observations, and consequently fail in scenarios where medium or sensor induces view-inconsistent degradations.

The primary source of failure under view-inconsistent degradations lies in the lack of physically-aware principles to guide scene factorization. Scene modeling approaches, such as NeRF and 3DGS, adopt indiscriminate representations of the scene and enforce multi-view consistency through volume rendering. These methods are vulnerable in view-inconsistent scenarios, where degradations cannot be interpreted as stable geometry or appearance. Specifically, participating medium can attenuate surface radiance, while sensor artifacts introduce stochastic fluctuations. In these cases, indiscriminate modeling without physically-aware factorization neglects the distinct properties of objects, medium, and sensors, often producing distorted geometry, manifested as floating artifacts and depth biases, and unreliable appearance, characterized by meaningless colors and texture shifts as shown in Fig. 1.

In this work, we propose a general framework, termed *Factorized Gaussian Splatting (F-GS)*, for modeling scenes under view-inconsistent degradations. We begin with a path-dependent image formation that explicitly decomposes radiative transfer into surface radiance, medium contributions, and residual sensor noise, thereby enabling physically-aware modeling of light attenuation along the ray. Building upon this formulation, the scene is factorized into three complementary components: **Geometry**, corresponding to surface radiance, represented by compact and high-opacity Gaussians modeling object surfaces; **Medium**, corresponding to medium contributions, represented by large, fixed, and low-opacity Gaussians approximating smooth attenuation and in-scatter; and **Residual**,

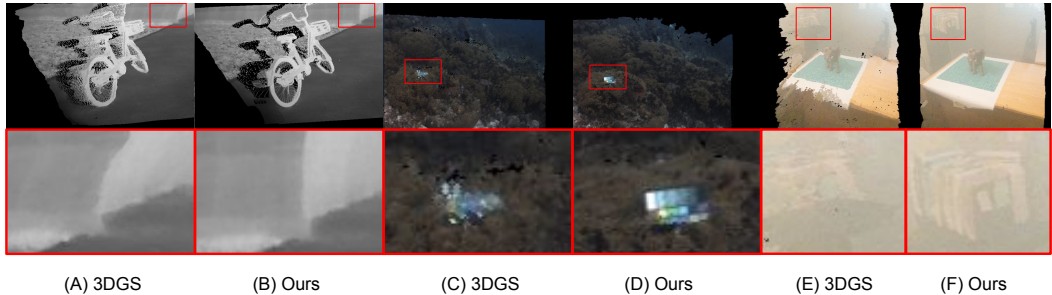

|  |  |  |  |  |  |
|---|---|---|---|---|---|
| (A) 3DGS | (B) Ours | (C) 3DGS | (D) Ours | (E) 3DGS | (F) Ours |

Figure 1: Qualitative reprojection comparison of our **F-GS** vs. **3DGS** under view-inconsistent degradations: **(A–B)** thermal LWIR, **(C–D)** underwater, **(E–F)** fog. We extract the alpha-blended depth from each model and visualize the depth points using ground truth colors from an alternative view. Our F-GS consistently suppresses depth biases and texture shifts compared to 3DGS.

corresponding to sensor noise and view-dependent artifacts, represented by small and low-opacity Gaussians together with a image-plane bias for fixed-pattern noise. To further stabilize geometry modeling, a lightweight *view-aware surface gain* and an optional *normal consistency* prior are further imposed on the geometry component. This factorization explicitly models medium and residual effects, thereby mitigating view-inconsistent degradations and producing sharper surfaces with stable radiance. Our main contributions are as follows:

- We introduce a physically-grounded and path-aware image formation, and propose Factorized Gaussian Splatting (F-GS) that disentangles the scene as Geometry, Medium, and Residual to isolate attenuation, path-radiance, and sensor artifacts.

- We stabilize geometry by coupling bounded-opacity Gaussian with a view-aware gain and, optionally, screen-space normal consistency and a small image-plane bias.

- We demonstrate that this factorization improves geometry fidelity and NVS quality across 3 important domains - thermal, fog and underwater, outperforming prior 3DGS pipelines.

## 2 RELATED WORK

### 2.1 MULTI-VIEW RECONSTRUCTION

Classical 3D reconstruction from multi-view images has traditionally relied on the Structure-from-Motion (SfM) Schönberger & Frahm (2016) and Multi-View Stereo (MVS) Schönberger et al. (2016) pipelines. Conventional SfM approaches estimate camera poses and recover sparse 3D structure by performing feature matching and triangulation across multiple views. MVS methods then refine this reconstruction by establishing dense pixel correspondences, typically using patch-based priors, to produce detailed geometric models. A recent trend in 3D reconstruction involves the adoption of volume rendering, which has significantly reshaped the domain of dense reconstruction. NeRF Mildenhall et al. (2020) introduced a neural rendering framework for learning radiance fields as implicit representations tailored for novel view synthesis, while NeuS Wang et al. (2021) extended this paradigm to support accurate multi-view geometry reconstruction. Building on the principles of volume rendering, 3DGS Kerbl et al. (2023) represents scenes as a collection of 3D Gaussians, enabling efficient optimization and real-time rendering through a carefully designed rasterization pipeline. Subsequent work has extended 3DGS to dynamic scenes Wu et al. (2024); Lu et al. (2024b), introduced advanced densification strategies Kim et al. (2024); Kheradmand et al. (2024); Li et al. (2024); Fang & Wang (2024a), and developed acceleration schemes Fan et al. (2024); Fang & Wang (2024b); Chen et al. (2024b); Charatan et al. (2024). In parallel, a growing body of research Guédon & Lepetit (2024); Huang et al. (2024); Yu et al. (2024) has focused on extracting surface meshes through multi-view reconstruction based on 3DGS.

## 2.2 Participating Medium in Neural Rendering

Participating medium (atmosphere, fog, turbid water) violate the cross-view appearance consistency assumed by standard neural rendering. Along each camera ray, direct radiance is exponentially attenuated and mixed with path radiance (backscatter/veiling), producing depth-dependent, view-inconsistent degradations that confound geometry estimation and radiance disentanglement.

Early efforts extend NeRF by incorporating attenuation and backscattering effects to enable haze, fog, and underwater synthesis, as well as dehazing applications Chen et al. (2023); Ramazzina et al. (2023); Levy et al. (2023). Building on Gaussian splatting, several variants have been proposed: WaterSplatting Li et al. (2025), which combines explicit Gaussian-splat-based geometry with per-pixel volumetric modeling; DehazeGS Yu et al. (2025), designed for multi-view haze removal; and SeaSplat Yang et al. (2025), targeting real-time underwater rendering. More recently, UDR-GS Du et al. (2024) extends this paradigm to dynamic underwater scenes by leveraging 4DGS Wu et al. (2023). While these works target scattering and attenuation, another important source of view-inconsistent degradation arises from thermal imaging through atmosphere.

Thermal computer vision has garnered growing interest in recent years, driven by the need for robust perception under adverse illumination and weather conditions. Recently, a handful of works have begun to apply volume rendering techniques to thermal data. Several studies Ye et al. (2024); Lin et al. (2024) adapt Neural Radiance Fields (NeRF) Mildenhall et al. (2020) to represent thermal scenes as implicit neural representations. Thermal3DGS Chen et al. (2024a) extends 3D Gaussian Splatting Kerbl et al. (2023) to the thermal domain by considering physical and camera effects. Other approaches tackle dynamic thermal sequences Liu et al. (2025) or integrate RGB and thermal modalities to leverage visible-light priors Hassan et al. (2024); Lu et al. (2024a). Existing approaches to thermal novel view synthesis predominantly emphasize image quality, whereas, in robotics applications employing thermal sensors, the primary requirement is accurate geometry. Departing from this trend, our method explicitly accounts for view-inconsistent degradations across observations, thereby facilitating robust geometric reconstruction in addition to high-quality rendering.

## 3 Methodology

### 3.1 Factorized Scene Formation

**Preliminary.** Gaussian representations model the scene as a collection of 3D Gaussians $\{\mathcal{G}_i \mid i = 1, \ldots, N\}$. Each Gaussian $\mathcal{G}_i$ is parameterized by its center $\boldsymbol{\mu}_i \in \mathbb{R}^3$, an opacity $\alpha_i \in [0, 1]$, and a covariance $\boldsymbol{\Sigma}_i = \mathbf{R}_i \, \text{diag}(\boldsymbol{\sigma}_i^2) \, \mathbf{R}_i^\top$, where $\boldsymbol{\sigma}_i \in \mathbb{R}^3$ are axis-aligned standard deviations and $\mathbf{R}_i$ is a rotation matrix derived from a quaternion $\mathbf{q}_i$. We adopt 3D Gaussians as the fundamental primitives in our pipeline due to their efficiency in scene recovery and ability to support high-fidelity rendering.

View-inconsistent scenarios are cases where the sensor-captured radiance of the same surface point varies across different viewpoints. Such inconsistency introduces significant challenges for scene reconstruction and typically manifests as artifacts such as ghosting, flickering, blurring, or distorted geometry. In the view-inconsistent setting, 3DGS indiscriminately represents the scene with Gaussian primitives and enforces multi-view consistency through volume rendering. To model view-dependent radiance variations, such as those introduced by highly reflective surfaces or camera-induced effects, 3DGS leverages spherical harmonics. This approach is effective for scenarios that do not involve participating medium (*e.g.*, water in underwater scans) or sensor noise (*e.g.*, low-light imaging with a non-ideal camera), such as standard RGB image inputs.

**Problem Formulation.** Given a set of $N$ images $\mathcal{I} = \{I_i\}_{i=1}^N$ captured by sensors with known intrinsic and extrinsic parameters under view-inconsistent degradations arising from participating medium and sensor effects, our pipeline is designed to model scenes in these degraded scenarios, simultaneously learning geometry and appearance.

A major challenge in modeling view-inconsistent degradations is that scene representations tend to entangle view-dependent effects originating from participating medium and sensor noise. Under real-world degraded conditions, the captured signal $I_i$ from a non-ideal sensor constitutes a path-dependent mixture of radiance originating from both objects and the medium, each with distinct physical properties. Accordingly, the measurement $I_i$ can be expressed as a path-dependent combination of

1) surface radiance, 2) medium contributions (*e.g.*, attenuation, backscatter/veiling, and forward-scatter blur that grows with path length and turbidity or aerosol load), and 3) residual noise (*e.g.*, temporal drift, photo-response non-uniformity, fixed-pattern noise). These components vary across viewpoints due to geometry-, path-, and time-dependent factors, and the use of spherical harmonics within standard Gaussian representations is insufficient to capture such effects. Consequently, vanilla Gaussian representations tend to absorb medium and sensor contributions into the scene model, often resulting in unstable modeling characterized by distorted geometry and unreliable appearance.

**Path-Aware Image Formation.**    To address the aforementioned view-inconsistent degradations, we propose a path-aware image formation as an alternative to the conventional volume rendering formulation Mildenhall et al. (2020), which relies on the assumption of strict view consistency. Our proposed physically informed path-aware formation decomposes each pixel measurement into attenuated surface radiance, accumulated medium radiance, and a residual term capturing forward-scatter blur and sensor noise. By explicitly modeling medium and sensor effects, it enables a clearer separation of geometry and appearance from degraded inputs.

Concretely, we propose our path-aware image formation model inspired by radiative transfer. Given a ray $r(s) = \mathbf{o} + s\mathbf{d}$ with origin $\mathbf{o}$ and direction $\mathbf{d}$, the observed radiance $I(\mathbf{x}, \omega)$ at pixel $\mathbf{x}$ and view direction $\omega$ can be expressed as:

$$I(\mathbf{x}, \omega) = T(d)\, S(\mathbf{x}, \omega) + \int_0^d T(s)\, \beta_s(s)\, J(s, \omega)\, ds + R, \qquad (1)$$

where $S(\mathbf{x}, \omega)$ is the surface radiance at the first surface hit at depth $d$, $\beta_s$ is the scattering coefficient, and $J(s, \omega)$ is the in-scattered source radiance. The transmittance can be formulated as:

$$T(s) = \exp\Big( -\int_0^s \beta_t(\mathbf{o} + u\mathbf{d})\, du \Big), \qquad \beta_t = \beta_a + \beta_s, \qquad (2)$$

where $\beta_a$ is absorption coefficient, and $\beta_s$ is scattering coefficient and $\beta_t$ is extinction coefficient. For simplicity, we adopt a single-scattering effective model for each spectral band $c$:

$$I_c(\mathbf{x}, \omega) \approx T_c(d)\, S_c(\mathbf{x}, \omega) \; + \; \big(1 - T_c(d)\big)\, M_c(\mathbf{x}, \omega) \; + \; R_c(\mathbf{x}, \omega), \qquad (3)$$

where $S_c$ models the surface radiance, $M_c$ captures medium contributions, and the residual term $R_c$ represents forward-scatter blur together with sensor noise.

## 3.2 Factorized Scene Modeling

Building upon our path-aware image formation, Factorized Gaussian Splatting (F-GS) decomposes the scene into three components: 1) a geometry component $S_c$ that encodes opaque surface appearance tied to the scene geometry (Sec. 3.2.1); 2) a medium component $M_c$ that models transmittance and in-scattered/path radiance along the viewing ray (Sec. 3.2.2); and 3) a residual component $R_c$ that captures view-dependent and sensor-locked artifacts not explained by the first two (Sec. 3.2.3). This factorization explicitly disentangles object surfaces, medium effects, and residual artifacts, thereby facilitating improved geometry and appearance modeling under view-inconsistent degradations.

### 3.2.1 Geometry Modeling

Vanilla 3DGS implicitly assumes that all view variations arise from stable surface radiance, so every Gaussian is interpreted as geometry. In adverse domains, however, view-inconsistent degradations are spuriously absorbed as geometric structure resulting in unstable reconstructions, such as floaters and fragmented depth, motivating us to constrain geometry to explain only the surface radiance term $S_c$ in Eq. 3 with a set of *Geometry Gaussians* $\mathcal{G}^{\mathrm{geo}} = \{(\mu_i, \theta_i, \alpha_i, \mathbf{c}_i)\}_i$, where $(\mu_i, \Sigma_i)$ are the 3D mean and anisotropic covariance, $\alpha_i$ is opacity, and $\mathbf{c}_i$ are spherical-harmonic (SH) color coefficients for view-dependent surface radiance. Gaussians are initialized from COLMAP SfM points. To stabilize geometry, we parameterize opacity with a bounded sigmoid:

$$\sigma(\theta; a, b) \;=\; a + (b - a)\, \mathrm{sigmoid}(\theta), \qquad \alpha_i^{\mathrm{geo}} \;=\; \sigma\big(\theta_i^{\mathrm{geo}}; \alpha_{\min}^{\mathrm{geo}}(t),\, 1\big), \qquad (4)$$

where $\alpha_{\min}^{\mathrm{geo}}(t)$ is annealed upward over iterations, promoting early translucency for coarse alignment and late saturation for sharp, opaque surfaces. Under front-to-back compositing, semi-transparent

geometry permits a pixel to be partially explained by both a thin front layer (*i.e.*, floaters) and the true surface behind, allowing floaters to persist. Enforcing high opacity induces an occlusion prior: a spurious foreground Gaussian would fully occlude the background and must alone account for the pixel color, which is penalized by the photometric loss. Consequently, such Gaussians are either pruned during training or converge onto the true surface.

**View-Aware Surface Gain.** Beyond the standard spherical-harmonic view-dependent color used in 3DGS, we introduce a lightweight view-aware surface gain applied to Geometry Gaussians. This module predicts a bounded multiplicative adjustment on the low-order color coefficients, conditioned on a low-frequency spatial code and a compact per-view embedding. Unlike the SH view-dependent color, an empirical low-order expansion of outgoing radiance over direction, our view-aware surface gain (VASG) absorbs view- and capture-specific photometric residuals, such as per-view exposure/white-balance drift, vignetting, sensor gain, and weak scene emission:

$$\tilde{c}_{i,0}(v) \;=\; g_{\mathrm{dc}}(\mu_i, v)\, c_{i,0}, \tag{5}$$

The gains are produced by an MLP that takes a low-frequency NeRF-like spatial encoding of the Geometry Gaussians location and a per-view embedding:

$$g_{\mathrm{dc}} \;=\; MLP\Big( \underbrace{\varphi(\mu_i)}_{\text{LF positional encoding}} \;,\; \underbrace{\mathbf{e}(v)}_{\text{per-view embedding}} \Big), \tag{6}$$

It serves as a photometric correction that improves robustness to per-view radiometric inconsistencies.

**Normal Consistency.** In regions with minimal color variation (*e.g.*, textureless surface such as matte walls and uniform ground), pure photometric fitting admits many explanations and tends to fragment geometry or let residuals "explain" shape. We therefore add a geometry-only screen-space depth–normal consistency prior Huang et al. (2024) that biases solutions toward contiguous, near-planar surfaces. Crucially, this prior is computed from the geometry Gaussians alone, medium and residual layers do not contribute, so it cannot be satisfied by haze/airlight or sensor noise and thus supplies floater-free geometric supervision. Among photometrically similar fits, the prior selects low-curvature, coherent geometry rather than residual-compensated fragments. In practice it complements our opacity annealing and residual opacity bound to assign low-frequency surfaces to the geometry layer. Let $w_i(\mathbf{p})$ be the *geometry-only* blending weights at pixel $\mathbf{p}$. The depth induced unit normal are $\mathbf{N}(\mathbf{p}) =$ computed form $D(\mathbf{p}) = \sum_{i \in \mathcal{G}^{\mathrm{geo}}} w_i(\mathbf{p})\, z_i$. where $z_i$ is the depth of Gaussian $i$. Let $\mathbf{n}_i$ denote the normal of splat $i$ (from its anisotropy or a local plane fit). We minimize the following alignment loss:

$$\mathcal{L}_{\mathrm{nc}} = \sum_{\mathbf{p}} \sum_i w_i(\mathbf{p}) \left(1 - \mathbf{n}_i^\top \mathbf{N}(\mathbf{p})\right). \tag{7}$$

The extra geometry-only rasterization adds modest overhead and may slightly lower image-space metrics by preferring geometric regularity; we enable it when accurate, complete geometry is prioritized over small photometric gains. Having restricted geometry to the $S_c$ in Eq. 3, the remaining view-inconsistent energy must be explained by the path-radiance $M_c$ accumulated along the ray and a residual term $R_c$. We next model these two components in the same Gaussian rendering framework.

### 3.2.2 MEDIUM MODELING

Haze, path radiance, and distance-dependent attenuation have no dedicated carrier in conventional splatting, so they are inadvertently baked into surface. Scattering and attenuation then bias depth and color, entangling medium behavior with geometry. We therefore introduce a dedicated medium layer whose rendering follows physical transmittance. To account for the path-radiance term $M_c$ and its attenuation $T_c$, we align Gaussian alpha compositing with physical transmittance. Consider a camera ray partitioned into segments $\{m\}$, with optical thickness $\Delta\kappa_m \approx \int_{t_{m-1}}^{t_m} \beta(r(s))\, ds$, and per-segment opacity $\alpha_m \triangleq 1 - \exp(-\Delta\kappa_m)$, yields the standard front-to-back transmittance after $M$ segments:

$$T_M \;=\; \prod_{m=1}^{M}(1 - \alpha_m) \;=\; \exp\Big(-\sum_{m=1}^{M}\Delta\kappa_m\Big) \;\xrightarrow[\Delta t \to 0]{}\; \exp\Big(-\int_0^d \beta(r(s))\, ds\Big), \tag{8}$$

which matches Eq. 2. With opacity encoding optical thickness, alpha compositing consistently discretizes transmittance, motivating medium Gaussians with small opacity and per-band path-radiance. We introduce a set of *Medium Gaussians* $\mathcal{G}^{\text{med}} = \big\{(\mu_j, \Sigma_j, \theta_j^{\text{med}}, \mathbf{c}_j)\big\}_{j=1}^{N_{\text{med}}}$, with 3D mean $\mu_j$, anisotropic covariance $\Sigma_j$, and a per-band SH path-radiance color vector $c_j$. To stabilize learning and enforce the *small optical-thickness* regime, we map raw parameters through a bounded sigmoid:

$$\alpha_j^{\text{med}} = \sigma\big(\theta_j^{\text{med}};\, \alpha_{\min}^{\text{med}}, \alpha_{\max}^{\text{med}}\big), \qquad 0 < \alpha_{\min}^{\text{med}} \le \alpha_{\max}^{\text{med}} \ll 1. \tag{9}$$

Medium Gaussians are initialized as large, sparse whose centers are uniformly sampled inside an *expanded* SfM bounding box. To reduce degeneracy with surfaces, we fix their centers during training and only learn scales, colors, and opacities.

### 3.2.3 RESIDUAL MODELING

Regions with minimal color variation (*i.e.*, textureless and ambiguous regions) are prone to overfitting: if every deviation is forced into surfaces, depth becomes unstable. Moreover, conventional splatting has no image-plane pathway for sensor-locked artifacts, causing them to leak into the 3D. We therefore introduce a residual channel that absorbs small discrepancies without competing for occlusion, implicitly preserving a sensible depth prior for geometry. The residual term in Eq. 3, denoted $R_c$, aggregates effects that are neither surface locked nor well explained by a thin participating medium-most notably forward-scatter blur that varies with path length and sensor-locked patterns (*e.g.*, PRNU/DSNU, banding). To model these in their appropriate domains, we decompose $R_c$ into two complementary parts, world-space residual Gaussians and image-plane bias.

**World-Space Residual Gaussians.** We assign weak and view-dependent perturbations that parallax with the scene to a set of *Residual Gaussians*. These Gaussians are initialized from input SfM points. To ensure they act only as small additive corrections and cannot compete with surfaces, we tightly bound their opacity as:

$$\alpha^{\text{res}} = \sigma\big(\theta^{\text{res}};\, 0, \alpha_{\max}^{\text{res}}\big), \qquad \alpha_{\max}^{\text{res}} \ll 1. \tag{10}$$

Separating these low-opacity splats from geometry is key to eliminating floaters while maintaining high-fidelity rendering. Residual Gaussians are composited in world space together with geometry and medium, contributing only small and view-dependent perturbations to the radiance.

**Image-Plane Bias.** Complementing the world-space residuals, we model sensor-locked, non-parallaxing artifacts with a *view-independent* additive field $R_{\text{fpn}} : \Omega \to \mathbb{R}$ with $\Omega = \{0, \ldots, H-1\} \times \{0, \ldots, W-1\}$. For pixel $(u, v) \in \Omega$,

$$R_{\text{fpn}}(u, v) = a_{\text{row}}[v] + a_{\text{col}}[u] + \sum_{k=0}^{K_h-1} \sum_{\ell=0}^{K_w-1} c_{k\ell}\, \phi_{k\ell}(u, v), \tag{11}$$

where $a_{\text{row}}[v] \in \mathbb{R}$ and $a_{\text{col}}[u] \in \mathbb{R}$ parameterize vertical/horizontal striping, and $c_{k\ell} \in \mathbb{R}$ are coefficients of the truncated low-frequency basis $\{\phi_{k\ell}\}$. The basis functions $\phi_{k\ell}$ are orthonormal 2D DCT-II, modes on $\Omega$, $\phi_{k\ell}(u, v) = \cos\big(\frac{\pi}{H}(u + \frac{1}{2})k\big)\cos\big(\frac{\pi}{W}(v + \frac{1}{2})\ell\big)$, so keeping only $K_h \times K_w$ low-frequency terms compactly captures slowly varying bias fields. Let $I_{\text{world}}(u, v)$ denote the world-space composite (geometry, medium, residual Gaussians). The final pixel is formulated as:

$$I_{\text{final}}(u, v) = I_{\text{world}}(u, v) + R_{\text{fpn}}(u, v), \tag{12}$$

assigning non-parallaxing structure to $R_{\text{fpn}}$ while view-dependent fluctuations remain in world space.

## 3.3 FACTORIZED SCENE RECOVERY

**Densification.** Building on 3DGS Kerbl et al. (2023) and 3DGS-MCMC Kheradmand et al. (2024), we use a lightweight *opacity-aware* scheme: per-splat opacity drives pruning and duplication (prune persistently low-opacity, low-visibility splats; duplicate high-opacity ones to add local capacity), avoiding maintaining Monte Carlo states. To keep the decomposition flexible, we periodically reset all *geometry* opacities to $0.5$ and add a tiny zero-mean jitter to them, preventing premature saturation and improving responsiveness to gradients. The same prune/duplicate rule applies to *residual* Gaussians; *medium* Gaussians are only pruned according to visibility to avoid degeneracy with surfaces.

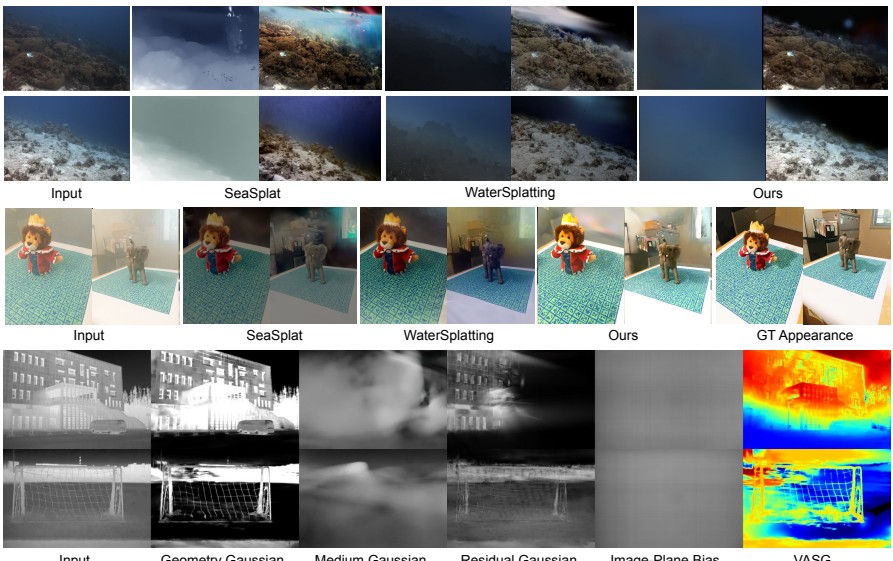

Figure 2: Qualitative visualization of Top: restoration on underwater scene. Middle: dehazing on fog scene. Bottom components visualization for F-GS in thermal where VASG are normalized in Jet.

**Optimization.** The three Gaussian families are composited front-to-back within the 3DGS rasterizer. Our objective combines photometric losses with regularization and (optionally) geometry priors:

$$\mathcal{L}_{\text{total}} = (1 - \lambda_{\text{D-SSIM}})\,\mathcal{L}_1 + \lambda_{\text{D-SSIM}}\,\mathcal{L}_{\text{D-SSIM}} + \mathcal{L}_\alpha + \mathcal{L}_{\text{scale}} + \lambda_{\text{NC}}\,\mathcal{L}_{\text{NC}}. \quad (13)$$

Here, $\mathcal{L}_1$ and $\mathcal{L}_{\text{D-SSIM}}$ measure photometric fidelity; $\mathcal{L}_\alpha$ and $\mathcal{L}_{\text{scale}}$ follow 3DGS-MCMC and regularize opacity and Gaussian extent; and $\mathcal{L}_{\text{NC}}$ is the normal consistency term that promotes contiguous, low-curvature geometry. The weights $\lambda_{\text{D-SSIM}}$ and $\lambda_{\text{NC}}$ balance perceptual and geometric regularity.

## 4 EXPERIMENTS

### 4.1 EXPERIMENTAL SETUP

**Datasets.** We evaluate F-GS on public benchmarks covering thermal, underwater, and foggy scenarios: **TI-NSD** Chen et al. (2024a) for thermal multi-view, **SeaThru-NeRF** Levy et al. (2023) for underwater scenes, and **DehazeNeRF** Chen et al. (2023) for fog. **TI-NSD** is a real-world, thermal-only dataset captured with 360° multi-view trajectories. We use scenes with valid COLMAP reconstructions from both the indoor and outdoor subsets, excluding scenes where pose recovery fails to ensure fairness. **SeaThru-NeRF** provides four real multi-view underwater scenes, each with approximately 30 RGB images. **DehazeNeRF** contains real indoor captures with hazy and clear images, about 60 hazy views and 80 clear views per scene. Only hazy images are used for experiments

**Implementation Details.** Following prior work, we reuse poses/initialization when available; images are downsampled to 400–800 px. Our F-GS is implemented on top of the gsplat rasterizer Ye et al. (2025). On an RTX 4090, F-GS trains 15 min per TI-NSD scene (24 min w. $\mathcal{L}_{\text{NC}}$), faster than Thermal3DGS (40 min) and moderately slower than MCMC (9 min). More implementation detail and parameters for densification strategy see appendix and repo.

**Metrics.** For novel-view synthesis, we report PSNR, SSIM, and LPIPS to assess rendering fidelity, following prior work by using every 8th image for evaluation. For geometry, we adopt a reprojection-error metric computed from rendered depth maps.

### 4.2 EXPERIMENTAL RESULTS

**Novel-View Synthesis.** Table 1 reports *quantitative* results on TI-NSD Chen et al. (2024a), comparing our method against Thermal3DGS Chen et al. (2024a), 3DGS, and 3DGS-MCMC. Our approach attains the best average PSNR and LPIPS, and ranks top-2 on SSIM across all scenes, indicating

superior fidelity and perceptual quality. MCMC achieves competitive (often best) SSIM on some scenes, consistent with its use of numerous low-opacity splats to fit fine detail and sensor noise, but this tends to trade off geometric regularity. Table 3 summarizes *quantitative* results on underwater and fog scenes. We compare againstWatersplatting Li et al. (2025), SeaSplat Yang et al. (2025), 3DGS, 3DGS-MCMC, and RestorGS Qiao et al. (2025). Our method attains the best or second best scores on nearly all sequences. Watersplatting, which uses an MLP to parameterize the participating medium, also performs strongly. In contrast, methods that model only attenuation (SeaSplat, RestorGS) trail in novel-view metrics, underscoring the benefit of explicitly modeling path radiance and residuals. Fig. 2 provide a visualization of the rendering including restoration, dehazing, and the factorization of each component.

**Geometry Evaluation.** We measure geometric consistency via a reprojection test. We render alpha blended depths and then back-project each valid pixel at depth $D(\mathbf{p})$ to 3D and re-project it into the image plane of the $k$-th preceding frame; The detailed definition is in the Appendix.

We report results on thermal sequences only: strong participating medium in fog/underwater scenes introduce path-dependent blur and airlight that violate the pinhole reprojection assumption, making correspondence unreliable. As shown in Table 2, our factorized, geometry-only depth yields the lowest errors for $k \in \{10, 15, 20\}$; adding the normal-consistency prior further reduces $\mathrm{RE}_k$, indicating crisper, more stable geometry. We also visualize the projected depth point in Fig. 3, results indicating our method generate crisper, more stable geometry.

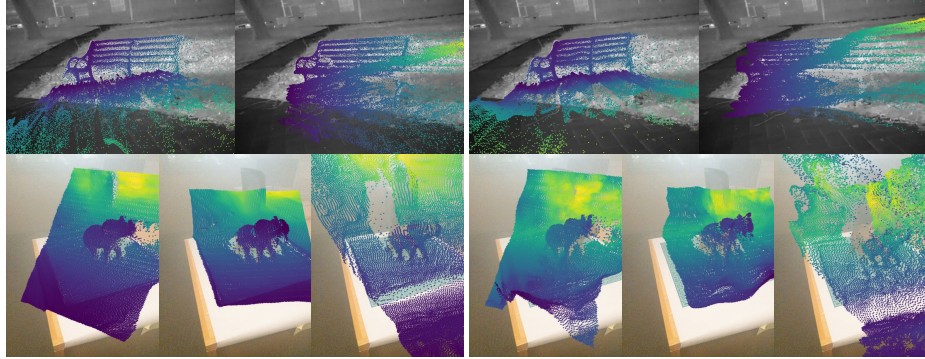

(A) Top: Ours;  Bottom: Ours + $\mathcal{L}_{nc}$          (B) Top: Thermal3DGS;  Bottom: MCMC

Figure 3: Qualitative depth projection analysis. We obtain alpha-blended depths from multi views and project the resulting depth points onto a reference frame. Color indicates estimated depth in plasma; Perfect projection should align precisely with the object. Misalignment between projected points and the silhouette indicates reconstruction errors.

Table 1: Quantitative thermal rendering comparison on the TI-NSD dataset. Higher is better for **PSNR** and **SSIM** (↑), and lower is better for **LPIPS** (↓). Top 2 methods are colored with Red and Orange.

| Metric | Method | Bicycle | Bridge | Car | Chair | Standing | Sitting | Goal | Building | Avg. |
|---|---|---|---|---|---|---|---|---|---|---|
| PSNR ↑ | 3DGS | 32.88 | 30.23 | 32.09 | 32.85 | 36.38 | 36.63 | 35.09 | 34.07 | 33.78 |
| | Thermal3DGS | 32.68 | 30.05 | 31.91 | 33.23 | 37.32 | 37.86 | 34.27 | 33.71 | 33.88 |
| | MCMC | 33.06 | 31.43 | 34.78 | 32.84 | 35.98 | 36.40 | 35.82 | 34.28 | 34.33 |
| | Ours.w.$\mathcal{L}_{nc}$ | 34.05 | 31.78 | 35.03 | 33.72 | 37.83 | 38.02 | 35.70 | 34.69 | 35.10 |
| | Ours | 34.27 | 32.06 | 35.15 | 33.88 | 36.97 | 38.18 | 35.78 | 34.79 | 35.13 |
| SSIM ↑ | 3DGS | 0.953 | 0.922 | 0.940 | 0.937 | 0.967 | 0.968 | 0.957 | 0.946 | 0.949 |
| | Thermal3DGS | 0.952 | 0.911 | 0.939 | 0.937 | 0.968 | 0.970 | 0.952 | 0.945 | 0.947 |
| | MCMC | 0.955 | 0.930 | 0.961 | 0.943 | 0.968 | 0.969 | 0.962 | 0.947 | 0.954 |
| | Ours.w.$\mathcal{L}_{nc}$ | 0.957 | 0.925 | 0.957 | 0.941 | 0.969 | 0.971 | 0.959 | 0.950 | 0.953 |
| | Ours | 0.958 | 0.926 | 0.958 | 0.941 | 0.968 | 0.971 | 0.959 | 0.950 | 0.954 |
| LPIPS ↓ | 3DGS | 0.165 | 0.200 | 0.198 | 0.164 | 0.219 | 0.209 | 0.171 | 0.143 | 0.184 |
| | Thermal3DGS | 0.175 | 0.217 | 0.202 | 0.171 | 0.221 | 0.213 | 0.192 | 0.152 | 0.193 |
| | MCMC | 0.177 | 0.206 | 0.177 | 0.165 | 0.234 | 0.228 | 0.164 | 0.149 | 0.187 |
| | Ours.w.$\mathcal{L}_{nc}$ | 0.169 | 0.197 | 0.177 | 0.163 | 0.223 | 0.215 | 0.159 | 0.136 | 0.180 |
| | Ours | 0.162 | 0.189 | 0.174 | 0.161 | 0.219 | 0.209 | 0.155 | 0.134 | 0.175 |

Table 2: Mean reprojection error to the $k$th preceding views ($\text{RE}_k \downarrow$) for each scene in TI-NSD.

| Metric | Method | Bicycle | Bridge | Car | Chair | Standing | Sitting | Goal | Building | Avg. |
|---|---|---|---|---|---|---|---|---|---|---|
| $\text{RE}_{20} \downarrow$ | 3DGS | 29.38 | 28.61 | 34.65 | 31.88 | 31.05 | 34.09 | 22.28 | 25.98 | 29.74 |
| | Thermal3DGS | 26.39 | 23.94 | 31.92 | 29.11 | 30.56 | 34.21 | 20.67 | 20.28 | 27.14 |
| | MCMC | 25.18 | 21.91 | 25.08 | 28.79 | 30.62 | 32.59 | 16.86 | 16.58 | 24.70 |
| | Ours | 24.32 | 21.50 | 23.22 | 28.25 | 27.68 | 30.67 | 17.32 | 15.77 | 23.59 |
| | Ours.w.$\mathcal{L}_{\text{nc}}$ | 22.57 | 19.96 | 22.34 | 26.50 | 27.40 | 29.92 | 16.04 | 14.49 | 22.40 |
| $\text{RE}_{15} \downarrow$ | 3DGS | 26.21 | 26.30 | 30.31 | 27.69 | 28.21 | 30.60 | 20.01 | 25.25 | 26.82 |
| | Thermal3DGS | 23.18 | 21.79 | 27.64 | 25.19 | 27.74 | 30.23 | 18.04 | 18.81 | 24.08 |
| | MCMC | 21.97 | 20.01 | 21.47 | 24.96 | 27.73 | 28.92 | 14.91 | 15.06 | 21.88 |
| | Ours | 21.55 | 19.76 | 20.10 | 24.75 | 25.28 | 28.59 | 15.16 | 14.41 | 21.20 |
| | Ours.w.$\mathcal{L}_{\text{nc}}$ | 19.59 | 18.33 | 19.30 | 23.05 | 24.93 | 27.49 | 14.12 | 13.26 | 20.01 |
| $\text{RE}_{10} \downarrow$ | 3DGS | 29.38 | 28.61 | 34.65 | 31.88 | 31.05 | 34.09 | 22.28 | 25.98 | 29.74 |
| | Thermal3DGS | 26.39 | 23.94 | 31.92 | 29.11 | 30.56 | 34.21 | 20.67 | 20.28 | 27.14 |
| | MCMC | 25.18 | 21.91 | 25.08 | 28.79 | 30.62 | 32.59 | 16.86 | 16.58 | 24.70 |
| | Ours | 24.32 | 21.50 | 23.22 | 28.25 | 27.68 | 30.67 | 17.32 | 15.77 | 23.59 |
| | Ours.w.$\mathcal{L}_{\text{nc}}$ | 16.10 | 15.81 | 15.10 | 18.45 | 21.58 | 22.40 | 11.79 | 11.17 | 16.55 |

Table 3: Quantitative comparison in underwater and fog. (metrics: PSNR ↑, SSIM ↑, LPIPS ↓).

| Method | IUI3 Red Sea | | | Curacao | | | J.G. Red Sea | | | Panama | | | Elephant(fog) | | | Lion(fog) | | |
|---|---|---|---|---|---|---|---|---|---|---|---|---|---|---|---|---|---|---|
| | PSNR | SSIM | LPIPS | PSNR | SSIM | LPIPS | PSNR | SSIM | LPIPS | PSNR | SSIM | LPIPS | PSNR | SSIM | LPIPS | PSNR | SSIM | LPIPS |
| 3DGS | 26.19 | 0.936 | 0.033 | 30.17 | 0.926 | 0.085 | 21.28 | 0.849 | 0.096 | 30.62 | 0.940 | 0.034 | 30.34 | 0.877 | 0.221 | 29.90 | 0.918 | 0.146 |
| MCMC | 31.13 | 0.957 | 0.017 | 30.43 | 0.933 | 0.061 | 23.05 | 0.879 | 0.067 | 31.77 | 0.949 | 0.030 | 33.64 | 0.891 | 0.191 | 32.90 | 0.931 | 0.121 |
| Watersplatting | 30.14 | 0.948 | 0.019 | 33.28 | 0.958 | 0.039 | 24.58 | 0.896 | 0.049 | 30.65 | 0.930 | 0.031 | 33.42 | 0.920 | 0.217 | 33.44 | 0.944 | 0.122 |
| Seasplat | 29.13 | 0.950 | 0.024 | 30.94 | 0.935 | 0.077 | 23.13 | 0.881 | 0.078 | 27.80 | 0.914 | 0.059 | 27.93 | 0.861 | 0.202 | 27.42 | 0.883 | 0.163 |
| RestorGS | 29.97 | 0.952 | 0.028 | 31.95 | 0.944 | 0.055 | 24.05 | 0.882 | 0.071 | 30.79 | 0.932 | 0.046 | - | - | - | - | - | - |
| Ours | 31.61 | 0.952 | 0.024 | 34.76 | 0.964 | 0.037 | 23.31 | 0.885 | 0.061 | 32.23 | 0.952 | 0.025 | 33.95 | 0.893 | 0.177 | 33.47 | 0.929 | 0.119 |

Table 4: Independent ablation on TI-NSD (thermal) and SeaThru-NeRF (underwater)

| | TI-NSD | | | | Seathru-NeRF | | |
|---|---|---|---|---|---|---|---|
| | PSNR↑ | SSIM↑ | LPIPS↓ | $RE_{20}$↓ | PSNR↑ | SSIM↑ | LPIPS↓ |
| Ours | 35.13 | 0.954 | 0.175 | 23.59 | 30.48 | 0.938 | 0.037 |
| w.o. View-aware surface gain | 34.83 | 0.953 | 0.184 | 23.75 | 29.87 | 0.936 | 0.038 |
| w.o. Image-plane bias | 34.64 | 0.952 | 0.179 | 23.65 | 30.41 | 0.937 | 0.037 |
| w.o. Residual Geometry factorization | 35.07 | 0.953 | 0.177 | 24.55 | 29.39 | 0.931 | 0.039 |
| w.o. Medium modeling | 35.11 | 0.954 | 0.177 | 24.02 | 29.76 | 0.935 | 0.039 |

**Ablation Study.** We perform an *independent* ablation on TI-NSD (thermal) and SeaThru-NeRF (underwater). Starting from the full model (**Ours**), we remove: (i) the *view-aware surface gain* (VASG) in geometry, (ii) the *view-independent image-plane bias*, (iii) the *residual/geometry factorization* (bounded opacities), and (iv) *medium modeling*. Removing the view-aware surface gain hurts perceptual quality the most, confirming it's what keeps appearance consistent across viewpoints. Skipping the image-plane bias causes small but reliable declines, suggesting it quietly soaks up sensor-locked artifacts without interfering with geometry. Turning off the residual/geometry factorization makes geometry less stable under reprojection, which shows that separating residuals from surfaces is key to clean depth. Dropping medium modeling matters little on thermal scenes but clearly degrades underwater performance, where path radiance and attenuation are stronger.

## 5 CONCLUSION

In this work, we present Factorized Gaussian Splatting (F-GS), a physically motivated scene modeling framework designed to handle view-inconsistent degradations arising from participating medium and sensor noise. Our framework decomposes a scene into three complementary components: Geometry, which consolidates scene structure while isolating it from degradations; Medium, which models light attenuation through participating medium; and Residual, which compensates for sensor-induced variations. By disentangling geometry, medium, and residual contributions, F-GS transforms real-world and view-inconsistently degraded inputs into a consistent scene representation. We evaluate F-GS across thermal, underwater, and foggy benchmarks. Our approach demonstrates substantial improvements in both novel-view synthesis quality and geometric stability compared to baseline methods. These results establish a significant step toward physically-aware scene modeling within the paradigm of 3D Gaussian Splatting.

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

# A APPENDIX

## A.1 IMPLEMENTATION DETAIL

F-GS is trained for 30K iterations on thermal and fog datasets using the Adam optimizer with the same base learning rate as 3DGS. For the SeathruNeRF dataset, we follow prior work and train for only 15K iterations. To ensure a fair comparison, we evaluate SeathruNeRF using the AlexNet-based LPIPS metric as in the unpublished RestorGS baseline, while all other experiments report the VGG-based LPIPS score consistent with 3DGS.

For residual Gaussians, we replace the default exponential learning-rate decay with a linear schedule that reduces the rate to 1% of its initial value, preventing over-decomposition of geometry. The number of primitives is set to 500K geometry Gaussians, 50K residual Gaussians, and medium Gaussians for TI-NSD; for underwater and fog scenes, which are smaller in scale, we use 250K geometry and residual Gaussians. We further apply opacity and scale regularization with coefficients of $1 \times 10^{-2}$ for thermal and fog datasets, and a lower $1 \times 10^{-3}$ for underwater. Full implementation details are provided in the code release at `https://anonymous.4open.science/r/F-GS-83EE/README.md`.

## A.2 DETAILS OF GEOMETRY EVALUATION

**Depth blending.** For geometry metrics we render a depth map but *only* over Geometry Gaussians $\mathcal{G}^{\text{geo}}$. With camera centre $\mathbf{o}$ and unit view direction $\mathbf{d}$, each pixel's ray is $\mathbf{r}(t) = \mathbf{o} + t\mathbf{d}$, $t \geq 0$. Depth at pixel $\mathbf{p}$ is the weighted expectation:

$$D(\mathbf{p}) = \sum_{i \in \mathcal{G}^{\text{geo}}} w_i(\mathbf{p}) z_i, \qquad z_i = \mathbf{d}^\top (\mu_i - \mathbf{o}),$$

where

$$w_i(\mathbf{p}) = \frac{T^{\text{geo}}_{i-1} \alpha_i \Pi_i(\mathbf{p})}{\sum_{j \in \mathcal{G}^{\text{geo}}} T^{\text{geo}}_{j-1} \alpha_j \Pi_j(\mathbf{p})}, \quad T^{\text{geo}}_{i-1} = \prod_{k < i, k \in \mathcal{G}^{\text{geo}}} (1 - \alpha_k).$$

**Photometric reprojection error.** For each frame $t$ and temporal lag $k$ we *lift* a pixel $\mathbf{u} = [u, v, 1]^\top$ to 3-D with its rendered depth $D_t(\mathbf{u})$ and the inverse intrinsics $K_t^{-1}$, transform the point into the camera of frame $t - k$, and project it back to the image plane. Denote this forward–backward mapping by $\mathbf{u} \mapsto \mathbf{u}_{t \to t-k}$. The per-pixel colour residual is then

$$e_{t,k}(\mathbf{u}) = \left\| I_t(\mathbf{u}) - I_{t-k}\big(\mathbf{u}_{t \to t-k}\big) \right\|_1,$$

computed only for correspondences whose projection lies inside the target image. Averaging over all valid pixels $\mathcal{V}_{t,k}$ in frame $t$ and over all frames that admit the offset yields

$$\text{RE}_k = \frac{1}{|\mathcal{T}_k|} \sum_{t \in \mathcal{T}_k} \frac{1}{|\mathcal{V}_{t,k}|} \sum_{\mathbf{u} \in \mathcal{V}_{t,k}} e_{t,k}(\mathbf{u}),$$

where $\mathcal{T}_k = \{ t \mid t - k \geq 0, \; |\mathcal{V}_{t,k}| > 0 \}$. For table readability, we report the reprojection error in range 0 to 255.

## A.3 THE USE OF LARGE LANGUAGE MODELS (LLMS)

We used LLMs exclusively for language polishing (grammar and style). No experimental design, data analysis, figure creation, or result interpretation was performed by the tool. The authors reviewed and approved all edits.

