# OpenReview forum: "Factorized Gaussian Splatting for View-Inconsistent Degradations"
_ICLR.cc/2026/Conference — ICLR 2026 Conference Withdrawn Submission_

### Official Review · Reviewer_udfe · 2025-10-30

**Soundness:** 2
**Presentation:** 2
**Contribution:** 2
**Rating:** 4
**Confidence:** 4

**Summary:**

The paper introduces Factorized Gaussian Splatting (F-GS), an extension of 3D Gaussian Splatting designed for scenes with participating media and degraded conditions such as thermal, underwater, and fog environments. It decomposes the rendered radiance into geometry, medium, and residual components using a path-aware image formation model inspired by radiative transfer. The method achieves improved reconstruction and geometric stability compared to existing baselines and includes detailed ablations and runtime analyses across multiple datasets.

**Strengths:**

1. The paper provides a clear physical motivation grounded in radiative transfer theory, resulting in an interpretable formulation that decomposes radiance into geometry, medium, and residual components.
2. The proposed practical modules are thoroughly ablated and empirically shown to be effective, and extensive experiments across multiple datasets demonstrate state-of-the-art performance.

**Weaknesses:**

1. The path-aware formulation is conceptually clear but not mathematically enforced, making the disentanglement heuristic and potentially unstable.
2. The robustness of the decomposition to hyperparameters, initialization, and noise is unkown.
3. The 2D image-plane residual breaks multi-view consistency and may hide geometry errors.

**Questions:**

1. How robust is the proposed factorization during optimization? In particular, are there cases where the geometry, medium, and residual components fail to remain distinct across different random seeds or scene conditions? I would consider increasing my score if the proposed Factorized Gaussian Splatting is demonstrated to be robust across seeds and scenes without requiring extensive hyperparameter tuning.

2. Regarding the paper’s claim on addressing view inconsistency, it would be valuable to include a more comprehensive discussion of other potential sources of view inconsistency, such as dynamic objects and occlusions, as examined in related work on distractor-free NeRFs (e.g., NeRF On-the-Go, NeRF-W) and recent 3DGS variants. Such a discussion would help clarify the scope and limitations of the proposed approach.

---

### Official Review · Reviewer_agr1 · 2025-10-31

**Soundness:** 3
**Presentation:** 3
**Contribution:** 2
**Rating:** 4
**Confidence:** 4

**Summary:**

This paper aims to address the challenge of optimizing 3D Gaussian Splatting (3DGS) when input images captured from different viewpoints exhibit significant inconsistencies. The main contribution lies in decomposing the 3DGS representation into three hierarchical levels, which respectively model the geometry, intermediate states, and fine-grained view-specific variations. The effectiveness of the proposed method is validated on thermal long-wave infrared, underwater, and foggy scene datasets.

**Strengths:**

1. The paper proposes to explicitly reconstruct the main geometric component of the scene and applies most geometry-related constraints at this level, which is a reasonable and well-motivated design choice.
2. The hierarchical decomposition of 3DGS is demonstrated through visualizations in figure 2, showing the results of different layers.
3. The paper is clearly written and easy to understand and follow.

**Weaknesses:**

1. The proposed method involves a large number of predefined parameters for controlling the opacity of 3DGS at different hierarchical levels. These parameters play an important role in the optimization of each level, but introducing too many hyperparameters in such performance-critical design choices may compromise the robustness and generalizability of the approach.
2. The strategy for modeling view-specific components using a neural network is not particularly novel, as similar ideas have been explored in prior works such as GOF [1]  and related methods.
3. Since the method focuses on handling inconsistencies across viewpoints, a comparison with Wild3DGS methods, [2] which also targets scenes with significant illumination variations across views, would be necessary and strengthen the empirical evaluation.

[1] Gaussian Opacity Fields: Efficient Adaptive Surface Reconstruction in Unbounded Scenes
[2] WildGaussians: 3D Gaussian Splatting in the Wild

**Questions:**

My main questions concern the robustness of the proposed method to hyperparameter choices and its direct comparison with the Wild3DGS methods. The authors are also encouraged to clarify the similarities and differences between their approach and Wild3DGS methods.

---

### Official Review · Reviewer_QYu5 · 2025-11-01

**Soundness:** 3
**Presentation:** 2
**Contribution:** 3
**Rating:** 4
**Confidence:** 3

**Summary:**

The paper proposes Factorized Gaussian Splatting (F-GS), a path-aware decomposition of scenes into Geometry, Medium, and Residual components to handle view-inconsistent degradations. It stabilizes geometry with bounded opacities and a view-aware surface gain, models the medium using sparse low-opacity Gaussians with fixed centers, and captures remaining artifacts via world-space residual Gaussians plus an image-plane bias for sensor-locked artifacts. Experiments on thermal, underwater, and fog datasets show improved novel-view rendering and better depth reprojections over baselines.

**Strengths:**

- the proposed method achieves state-of-the-art performance on both photometric and depth reprojection scores under all experimented settings: thermal, foggy, and underwater scenes.
- the ablation studies clearly shows the benefit of each proposed component under both thermal and underwater scenes.

**Weaknesses:**

- Figure 2 lacks clarity and organization. As far as I can tell, row 1 illustrates underwater restoration, row 2 shows fog dehazing plus novel view synthesis, and row 3 presents a component breakdown, effectively mixing three use cases in one panel. The caption omits key details: in the top row, it is unclear whether the two columns per method depict two different scenes or separate medium/geometry renderings; in the middle row, it is unclear whether “Ours” shows the full factorized rendering or geometry-only. I recommend splitting the figure into three subfigures: (a) underwater, (b) fog, (c) thermal, and explicitly stating for each what every row and column represents, including whether images are full renderings or individual components.
- Figure 3 clarity. The figure is confusing because the top/bottom rows mix different methods and appear to use different scenes/views, preventing a fair visual comparison. The point overlay is also too opaque, obscuring the underlying image and making alignment hard to judge. Please (i) use the same scene and view for each method, (ii) place methods side by side with consistent crops, and (iii) reduce overlay opacity so the background is visible, or include the raw image without the depth overlay alongside the overlay for reference.
- Fixing the medium Gaussian centers means the haze can’t shift where it should, which can potentially cause blurry halos or artifacts, or suffer from inaccurate SfM initializations. Allowing small, controlled movement might be more flexible.

**Questions:**

Apart from the issues mentioned in the weaknesses, I wonder if the ablation studies could also be performed on foggy scenes for completeness, since the other two scenerios are covered. I also recommend the authors to provide some side-by-side video comparisons of the proposed method, baselines, and ground truth along moving camera trajectories. These clips would make the rendering quality much easier to assess than static frames alone.

---

### Official Review · Reviewer_BJ6s · 2025-11-01

**Soundness:** 2
**Presentation:** 2
**Contribution:** 2
**Rating:** 2
**Confidence:** 4

**Summary:**

To model the multi-view reconstruction in the presence of participating medium and sensor artifacts, this paper proposes Factorized Gaussian Splatting that decomposes the scene into three components: geometry, medium, and residual. Specifically, each component is modeled with separate Gaussians. Moreover, to obtain a stable geometry component, the view-aware gain and the normal-consistency prior are utilized to enhance the scene structure. Experiments are conducted in thermal, underwater, and foggy scenes.

**Strengths:**

- This submission provides the anonymous code, which merits encouragement.

- This paper achieves a physically-aware modeling of complex scenes via the decomposition of object surfaces, medium effects, and residual artifacts.

**Weaknesses:**

- The step from Equation 1 to 3 is justified not only by single scattering but also implicitly requires (i) an effectively homogeneous (or slowly varying) medium along each ray so that $T(d)=e^{-\beta_t d}$ and $\beta_s J$ can be treated as path-constant, (ii) a fixed treatment of the phase function folded into $J$.

- The training appears highly sensitive to the annealing schedule in Equation 4: increasing $\alpha_{\min}$ too early can lock in incorrect geometry, whereas increasing it too late makes it difficult to remove floaters. The manuscript lacks a clear schedule and an ablation study for this part.

- The physical meaning of $M_c(\mathbf{x},\omega)$ is unclear. As written, for it appears to play the role of a **global atmospheric light term** in hazy scenes. Under this interpretation, Equation 3 reduces to the standard airlight formulation and is essentially the same as DehazeGS, except for the explicit residual term.

- Many hyper-parameters in Equations 4, 9–10 are not provided in this paper, and it is also difficult to locate them in the provided code. These hyper-parameters are very important for verifying the generalization of the proposed modeling framework.

- The performance improvement of the proposed method, compared to MCMC, appears to be limited, especially given that the proposed method takes a longer training time (24 min) than MCMC (9 min).

- Table 3 is incomplete (no bottom line). Furthermore, there is no visual illustration of the proposed method; this is not common for a CV-related submission.

**Questions:**

Please see the weakness for detailed questions. My rating actually lies in the middle of scores 2 and 4, and the authors are encouraged to provide rebuttals.

---

### Note · Authors · 2025-12-01

**Comment:**

We sincerely thank the efforts from all reviewers. We have decided to substantially improve the manuscript and run additional experiments before submitting to another venue.

**Withdrawal Confirmation:**

I have read and agree with the venue's withdrawal policy on behalf of myself and my co-authors.